# A Neuroergonomic Approach Fostered by Wearable EEG for the Multimodal Assessment of Drivers Trainees

**DOI:** 10.3390/s23208389

**Published:** 2023-10-11

**Authors:** Gianluca Di Flumeri, Andrea Giorgi, Daniele Germano, Vincenzo Ronca, Alessia Vozzi, Gianluca Borghini, Luca Tamborra, Ilaria Simonetti, Rossella Capotorto, Silvia Ferrara, Nicolina Sciaraffa, Fabio Babiloni, Pietro Aricò

**Affiliations:** 1Laboratory of Industrial Neuroscience, Department of Molecular Medicine, Sapienza University of Rome, 00185 Rome, Italy; daniele.germano@uniroma1.it (D.G.); gianluca.borghini@uniroma1.it (G.B.); capotorto.1843967@studenti.uniroma1.it (R.C.); fabio.babiloni@uniroma1.it (F.B.); 2BrainSigns srl, 00198 Rome, Italy; andrea.giorgi@uniroma1.it (A.G.); vincenzo.ronca@uniroma1.it (V.R.); alessia.vozzi@uniroma1.it (A.V.); luca.tamborra@uniroma1.it (L.T.); ilaria.simonetti@uniroma1.it (I.S.); silvia.ferrara@brainsigns.com (S.F.); nicolina.sciaraffa@gmail.com (N.S.); pietro.arico@uniroma1.it (P.A.); 3Department of Anatomical, Histological, Forensic and Orthopaedic Sciences, Sapienza University of Rome, 00185 Rome, Italy; 4Department of Computer, Control, and Management Engineering Antonio Ruberti, Sapienza University of Rome, 00185 Rome, Italy; 5School of Computer Science and Technology, Hangzhou Dianzi University, Hangzhou 310018, China

**Keywords:** neuroergonomics, wearable EEG, Passive Brain–Computer Interface (BCI), mental effort, driving education, learning, road safety, car simulator

## Abstract

When assessing trainees’ progresses during a driving training program, instructors can only rely on the evaluation of a trainee’s explicit behavior and their performance, without having any insight about the training effects at a cognitive level. However, being able to drive does not imply knowing how to drive safely in a complex scenario such as the road traffic. Indeed, the latter point involves mental aspects, such as the ability to manage and allocate one’s mental effort appropriately, which are difficult to assess objectively. In this scenario, this study investigates the validity of deploying an electroencephalographic neurometric of mental effort, obtained through a wearable electroencephalographic device, to improve the assessment of the trainee. The study engaged 22 young people, without or with limited driving experience. They were asked to drive along five different but similar urban routes, while their brain activity was recorded through electroencephalography. Moreover, driving performance, subjective and reaction times measures were collected for a multimodal analysis. In terms of subjective and performance measures, no driving improvement could be detected either through the driver’s subjective measures or through their driving performance. On the other side, through the electroencephalographic neurometric of mental effort, it was possible to catch their improvement in terms of mental performance, with a decrease in experienced mental demand after three repetitions of the driving training tasks. These results were confirmed by the analysis of reaction times, that significantly improved from the third repetition as well. Therefore, being able to measure when a task is less mentally demanding, and so more automatic, allows to deduce the degree of users training, becoming capable of handling additional tasks and reacting to unexpected events.

## 1. Introduction

Car driving is considered a very complex activity, consisting of different concomitant tasks and subtasks that require a very high mental and physical coordination [1,2]. Within human factors research, a distinction has been made between ‘driver performance’, which reflects what a driver can do, based on his or her physical and mental capabilities, and ‘driver behavior’, which involves what a driver actually does and is influenced by social factors, cognitive and emotional phenomena, and self-motivations [3]. Despite the conceptual difference, it is often hard to distinguish driver behavior and performance, because of a strong inter-relation. Moreover, this inter-relation is often blind to an external observer, therefore it is crucial to pursue research on driver behavior and the reasons underlying certain risky attitudes behind the wheel.

This conflict is already present, and to a greater degree, while learning to drive. Most people who register for a driving school do so in order to obtain a driving license, not to learn how to drive properly and safely. The result is a relative easiness in obtaining a driver’s license, but it is not necessarily true that being able to drive (thus licensed) corresponds to knowing how to drive properly and safely [4]. Not surprisingly, the highest percentage of drivers injured as a result of a traffic accident is found in the 20–24 age group, as reported by an Italian Government investigation (Figure 1, source: ISTAT 2019, the last report without the interference of restrictions due to COVID-19 pandemic) [5], and similarly confirmed also in other countries, at least in Europe [6]. Besides these reports, there is no doubt about the increased risk associated with young drivers worldwide [7], and the relevance of appropriate driving education to mitigate such concern.

Driving education should therefore be a major social issue, with the government concerned with ensuring that current protocols are able to guarantee adequate levels of road safety, by taking into account new urban needs and dynamics, a new kind of highly varied and promiscuous mobility, and the technological evolution of vehicles. Instead, incredibly, driving education does not receive the attention it deserves; it is still tied to old and unregulated practices, thus resulting in an obsolete public service. Not surprisingly, the European Commission is currently preparing a major revision to the EU Driving License Directive (2006/126), with the main objective of enhancing road safety, as reported by the European Transport Safety Council [8], so contributing to ‘Vision Zero’ and the targets of reducing road deaths and serious injuries by 50% by 2030. At the same time, the European Commission is interested in harmonizing and standardizing driving license and related education across Europe [9].

In the current scenario, driving school trainees have to attend a certain number of driving practical sessions to achieve the proper level of knowledge, dexterity, and experience. The number of sessions in general depends on national regulations, with a certain degree of variability across Europe. Moreover, at a national level, the programs are almost standardized, being based on the repetition of specific driving tasks until the instructor is satisfied with the trainee’s driving performance. Therefore, trainees’ assessment mainly relies on the instructor’s subjective evaluation, with all the related concerns. In the “best” case, an inaccurate evaluation of the trainee’s driving abilities will lead to their rejection during the driving test, thus mainly harming the trainee themselves, who will have to incur new financial and time costs to repeat the course. In the worst case, the trainee will be dexterous enough to pass the test and obtain the license, but it could still be not ready (from a psychological point of view) for driving safely in the urban traffic. Knowing how to control a vehicle does not imply knowing how to drive safely in a real context, where several external variables and unexpected events can alter the mental effort required by the driver. It has been demonstrated that driving a car has not to be intended as a single task but as a combination of multiple tasks, with a certain level of hierarchy [2]. In some conditions, such as with high traffic, in a complex road infrastructure, and with bad weather conditions, the mental demand can become very high, thus “saturating” the driver’s mental resources and making him/her unable to react to sudden and unexpected events [10]. For an external evaluator, such as the driving instructor, it is difficult to actually understand the amount of cognitive resources employed by a driver, thus its assessment will be mainly based on the driver’s explicit behavior but not on its mental abilities.

To this regard, new effective solutions may be provided by the emerging field of the applied neuroscience and neuroergonomics [11,12,13,14,15]. It has been widely demonstrated how the measurement of human neurophysiological activities, such as electroencephalography (EEG), electroculography (EOG), electrodermal activity (EDA), and hearing activity by means of photopletismography (PPG), could allow to obtain useful insights and objective information about the cognitive and emotional dimensions of user’s experience [16,17,18,19,20]. In addition, the recent technological progress is leading to the diffusion of light, wearable, minimally invasives and reliable monitoring devices, both in terms of headset for EEG [21,22,23] and wristbands and other sensors for autonomic signals [24,25,26], therefore, there are no doubts about a larger and larger deployment of such devices for out-of-the-lab applications in the near future. 

In terms of neurophysiological monitoring, the EEG technology is considered the gold standard for the investigation of user’s mental states during a certain (operational) activity, the so-called passive Brain–Computer Interface applications [27,28], being a direct measure of Central Nervous System and so of the brain activity (while EOG, EDA, and PPG data are related to the Autonomous Peripheral Nervous System) with high temporal resolution (in the order of milliseconds), sensitivity, and wearability [25,26,29,30,31,32]. The EEG in particular is successfully deployed in scientific research applied to the automotive field for investigating the cognitive phenomena arising into the driver’s brain, mainly the mental effort with respect to different driving conditions and situations [33,34,35,36], attentional demand, allocation and related concerns [37,38,39,40], and the large and hot-topic area of mental fatigue, with the related concepts of drowsiness and sleepiness [41,42,43,44,45]. However, all the previously mentioned manuscripts employing EEG technology to measure user’s mental states were focused on applications other than driving education, even if still in the automotive field. For instance, Kim and colleagues [35] analyzed the variations in brain activity, related to mental workload, depending on the kind of road section in an urban environment, while Di Flumeri and colleagues [33] used the EEG-based workload measure to investigate the different impact of traffic conditions and road type. Another approach is to increase the sensitivity to the cognitive phenomenon by fusing EEG data with vehicular and contextual ones, as carried out by Islam and colleagues [36] for evaluating driver’s workload in a urban context, and by Guo and colleagues [37] to investigate driver’s vigilance and situation awareness. Therefore, there are no doubts about the potential of deploying EEG-based driver’s monitoring systems [46]; however, its use in driving education is still almost unexplored, apart from a few; however, these are not directly focused on assessing trainee’s mental improvements [47]. 

In any case, the different investigated mental states, such as mental effort, attention, and stress, could be measured through a specific “neurometric”, i.e., a synthetic and unique measure that is computed as the result of one or more neurophysiological features [48]. 

In this context, the rationale of the present study is to investigate whether the use of an EEG-based neurometric of the trainee’s mental effort during a learning-to-drive course can provide objective information, more comprehensive than the traditional supervisor observations, about the cognitive progress the student is making. A tool capable to provide an objective measure of the driving trainee’s mental workload deployed while driving would become a powerful solution for the instructor to obtain a complete overview of the actual trainee’s readiness to drive.

To achieve this objective, 22 young people, without or with limited driving experience, have been engaged in an experimental protocol based on a driving simulator. They were asked to drive along five different urban routes, but similar in terms of difficulty, while their brain activity was recorded through a wearable and minimally invasive EEG system. Moreover, driving performance, subjective and reaction times measures were collected.

Hereinbelow, the manuscript will be organized in the following structure:Section 2 is aimed at providing a clear and detailed description of the overall methodological implant, including the experimental design and related tasks, the data collection and processing, and the following statistical analysis;Section 3 is aimed at describing the obtained results;Section 4 is aimed at discussing results, even commenting the weaknesses of the present study and suggesting further improvements;Section 5, being the conclusions, is aimed at summarizing the main outcomes of the study.

## 2. Materials and Methods

The present study stems from the need to investigate potential innovative tools for assessing a trainee’s degree of learning, taking into consideration not only the overt behavioral aspects, but also and especially the cognitive processes from which these behaviors arise.

In particular, the study aimed at investigating whether the use of an EEG-based neurometric of the trainee’s mental effort during a learning-to-drive course can provide objective information, more comprehensive than the traditional supervisor observations, about the cognitive progress the student is making.

The experiments took place at the University of Burgos (Spain) and lasted almost 2 h per participant. One of the inclusion criteria was the absence of driving experience, since the main experimental objective was to validate the capability of assessing the trainees’ mental improvements while learning to drive. Therefore, the experiments were performed directly at the university in order to simplify the engagement of young adults without a driving license until that time. Two car simulators were set up in an artificially lighted room, as shown in the picture below (Figure 2). Each car simulator consisted of a realistic car cockpit, including a real car seat with safety belt, a pedal set, a manual gearbox, and a steering wheel with various actuators (directional indicators, lights, horn, etc.), and a three-screens display, in order to promote immersivity. Audio was provided through a headset, in order to isolate the participants from eventual acoustic interferences of the surroundings. During all the experimental tasks the participants’ brain activity was recorded through the Mindtooth Touch EEG wearable system (Brain Products GmbH. Gilching, Germany—BrainSigns srl. Rome, Italy). Such an EEG system, developed along the Horizon2020 European Fast-Track-to-Innovation Project Mindtooth (www.mindtooth.com, accessed on 10 October 2023, and https://mindtooth-eeg.com/ accessed on 10 October 2023), has been already successfully employed and validated in previous studies [49]. 

The following paragraphs will detail the experiments, the experimental sample, the designed tasks, and the employed technologies for data collection.

### 2.1. Experimental Task

Four different driving tasks were designed, and hereinafter described.

Some of the following tasks included a *failure* event. This failure consisted of the engine malfunction indicator light coming on, together with an acoustic alarm, on the car’s dashboard. The driver was instructed to press a specific button on the dashboard to switch off this alarm as soon as it realized it. The reaction time, in milliseconds, was gathered. These events were included within each scenario in a position-dependent way (i.e., they always happened in the same route position for all the participants), in order to allow a consistent comparison between participants. The presence of the *failure* event is detailed for each scenario. 

#### 2.1.1. Adaptation Task

A scenario within the same urban environment employed for the following driving tasks was designed to allow the participants to take confidence with the simulator, the car behaviors within the simulated environment and the GPS indicating the route to follow (Figure 3). The practice task did not include urban traffic, in order to avoid any initial effect of driving trainees’ training. This scenario was the same for all the participants and it lasted almost 6 min, depending on the driver speed (the drivers were instructed to respect the speed limit of 40 km/h, indicated by traditional road signage). Two *failures* were included within the task to train the participant to react to these events.

#### 2.1.2. Easy and Hard Driving Tasks

*Easy* and *Hard* tasks, performed on a test circuit outside the urban environment, aimed at inducing two “labelled” levels of driving mental workload (i.e., Easy and Hard) in order to preliminary validate the sensitivity of the workload EEG neurometric to the investigated phenomenon. 

The Easy task consisted of driving along a very easy circuit with an elliptical path at a fixed slow speed, without any traffic inside. The Hard task consisted of driving along a more complex circuit, with 90° turns, roundabouts, and slaloms between cones, with other traffic circulating inside the circuit.

Both driving tasks lasted almost 150 s (2 min 30 s).

No *failures* were included within these two tasks.

#### 2.1.3. Urban Driving Tasks

Seven different routes were designed along the same urban environment employed for the adaptation task. Since each participant had to perform 5 repetitions of this driving task, the related routes were different in order to avoid any driving improvement due to memorization of the route rather than an actual improvement in driving skills. At the same time, the routes were similar in terms of length, type of road, number of intersections and roundabouts crossed, and turns required. Moreover, all the tasks included similar traffic conditions. In this way, the seven routes can be considered similar in terms of difficulty (Figure 4).

The participants performed the five tasks in a casual sequence of routes. In particular, the sequences of the driving routes were different for each participant, they were designed in order to have a homogeneous repetition of all the 7 routes, and a homogenous recurrence of the 7 routes across the 5 repetitions (*run* in the following).

All the 7 tasks included 2 *failure* events at different distances from the starting point, in order to avoid any expectation behavior.

### 2.2. Experimental Participants

Twenty-four healthy participants, all students from the University of Burgos, were recruited on a voluntary basis. The experiments were conducted following the principles outlined in the Declaration of Helsinki of 1975, as revised in 2000. The experiments have been approved on the 4th of August 2020 by the Ethical Committee of Sapienza University, as reported in the Institutional Review Board Statement. Informed consent was obtained from each participant on paper, after the study explanation, as well as the consent for recording and employing videographical material. All the data were pseudorandomized to prevent any association with the participants’ identities. Two participants left the study after the first tasks, one because of motion sickness and one because of discomfort produced by the EEG system (the participant suffers from a skin pathology not communicated during recruitment). Therefore, the final experimental sample consisted of 22 participants, 19.2 ± 1.2 years old, 19 males and 3 females, with no or very little driving experience. In particular, 17 participants had no driving license, 1 for less than 6 months, and 4 for less than 12 months.

Each participant was involved for two consecutive hours in the same day. 

### 2.3. Experimental Protocol

On arrival, participants were asked to wear the Mindtooth headset. Sponges of wet-based electrodes were firstly disinfected, by using a solution of water and iso-propanol (70%), and then partially soaked with 2% NaCl saline solution. The quality of the EEG signal was checked by measuring the impedance values of each EEG electrode with respect to the reference and ground. Water-based electrodes contact was considered good for impedance values below 100 kΩ [50]. 

The participants firstly performed the ‘Adaptation task’ (urban environment, no traffic) in order to take confidence with the simulator, the commands, the GPS, and the *failure* events. Then, they were asked to perform two resting states, in particular one minute with open and one minute with closed eyes, needed for estimating individual EEG parameters needed for data processing (please refer to Section 2.4.1). After that, the participants had to follow the daily protocol summarized in the Table 1. After each Urban driving task, the participants were requested to self-assess the experienced workload by digitally filling the NASA-Task Load IndeX (NASA-TLX, please refer to Section 2.4.2) questionnaire. 

### 2.4. Methods

#### 2.4.1. Electroencephalographic Data Recording and Analysis

Each participant wore the Mindtooth Touch EEG headset, with 5 prefrontal electrodes (AFz, AF3, AF4, AF7, AF8), 3 parietal electrodes (P3, P4, Pz), plus ground (left mastoid) and reference (right mastoid). EEG raw data were recorded, at a sampling frequency of 125 Hz, through the Mindtooth proprietary software (v.5.22) running on a laptop.

The EEG data were offline processed by using the MATLAB software (Mathworks Inc., Natick, MA, USA), in its R2022b version. First of all, a 50-Hz notch filter was applied to remove interferences due to main line power. The EEG data were also band-pass filtered by means of a 5th order Butterworth filter (low-pass filter cut-off frequency: 30 Hz, high-pass filter cut-off frequency: 2 Hz), in order to remove the continuous and low frequency components causing data drifting, as well as high frequencies (higher than 30 Hz), where interference from other physiological activities such as muscle activity is generally coupled. Subsequently, the Reblinca [51] algorithm was used to detect blink artifacts. Through this information, it was possible to deploy a multichannel Wiener Filter [52] to estimate the blink-related component and to remove it from the EEG data. Then, for other sources of artifacts, dedicated algorithms of the EEGLAB toolbox [53] were applied. In detail, the blink-free signal was divided into 1-s-long epochs and a threshold criterion was applied, i.e., the epochs with the signal amplitude exceeding ±80 mV (threshold) were labelled as “artifacts” [54]. At the end, the EEG epochs marked as “artifacts” were removed from the EEG dataset with the aim to have a clean EEG signal to perform the analyses. Such a data-processing pipeline has been demonstrated to be robust and reliable in passive Brain–Computer Interfaces applications [22,30,32].

At this point, the global field power (GFP) was calculated from the artifact-free EEG with a focus on the area of interest for the mental effort. In particular, as Area of Interest it was considered the theta band over the frontal channels (AFz, AF3, AF4, AF7, and AF8). In fact, it has already been demonstrated in several contexts that the brain electrical activities mainly linked to the mental effort are the theta EEG rhythms, in particular over the prefrontal cortex (PFC) [41,55,56], and in a less extent the parietal EEG rhythms over the posterior parietal cortex (PPC) regions [33,57,58]. However, parietal activity has been linked also to other mental phenomena such as stress and memorization [49,59], therefore, it was decided to consider only the PFC theta activity, because of its stronger link with mental effort [60]. The GFP was chosen because it describes brain EEG activity with the advantage of representing, in the time domain, the degree of synchronization of a specific cortical region of interest in a specific frequency band [61]. 

At this point, before computing the GFP, the EEG data from the frontal channels were individually filtered, according to the Individual alpha Frequency (IAF) value [56], in theta band, defined as [IAF – 6 ÷ IAF – 2] Hz [62]. Once filtered, the mental effort (ME) index was so computed as in the following formula:$$ME\;(t)={GFP}_{frontal,\;\vartheta }=\frac{1}{N}\;\sum_{i=1}^{N}x_{i,\;\vartheta }^{2}\;(t)$$
where *N* is the number of the considered channels (5, i.e., AFz, AF3, AF4, AF7, and AF8), and *x_i,ϑ_*(*t*) is the *i*-th EEG channel filtered in theta (*ϑ*) band. Having been the original signal windowed into 1-s-long epochs, the time resolution of the *ME* index was equal to 1 s.

#### 2.4.2. Subjective Assessment

In this protocol, the NASA-Task Load IndeX (NASA-TLX) was employed, as shown within the experimental protocol. The NASA-TLX questionnaire has been proposed by Hart and Staveland [63]. The questionnaire is organized in six different dimensions:*Mental demand:* How much mental and perceptual activity was required? Was the task easy or demanding, simple or complex?*Physical demand:* How much physical activity was required? Was the task easy or demanding, slack or strenuous?*Temporal demand:* How much time pressure did you feel due to the pace at which the tasks or task elements occurred? Was the pace slow or rapid?*Performance:* How successful were you in performing the task? How satisfied were you with your performance?*Effort:* How hard did you have to work (mentally and physically) to accomplish your level of performance?*Frustration:* How irritated, stressed, and annoyed versus content, relaxed, and complacent did you feel during the task?

Twenty-step bipolar scales are used to obtain ratings for these dimensions. A score from 0 to 100 is obtained on each scale. 

A subjective measure of ‘Workload’, has been obtained by averaging the six scores [64]. In addition, the ‘Effort’ and ‘Performance’ scales have been analyzed also singularly in order to evaluate the participant’s self-evaluation of the overall difficulty (the former) and the performance satisfaction (the latter).

#### 2.4.3. Behavioural Assessment

The driving behavior was evaluated from two perspectives:-Driving performance: the simulator collects the number of collisions between the driver’s car and (i) the other cars, (ii) the road signs, and (iii) the road infrastructure, such as the sidewalks. The total number of collisions for each participant for each *Run* was estimated. It was hypothesized an inverse correlation with performance: the better the participant became at driving, the fewer collisions it would have to make.-Mental performance: the reaction times to the *failure* events were collected for each participant for each *Run* and averaged along the *Run* itself (there were two occurrence). In fact, the reaction to failures can be considered as a secondary task. According to the scientific literature related to the dual tasks interference, even while driving [65], with human mental resources being not unlimited, the more resources are allocated to the primary task, the more difficult it will be to manage a secondary task. Vice versa, the more experienced we become and able to perform the primary task automatically, the more we will improve our ability to handle secondary tasks, such as the reaction to additional and unexpected events. Therefore, a direct correlation was expected between improvement on the primary (driving) and the secondary (reaction to failure) tasks: the better the participant became at driving, the faster reaction times it would have to achieve.

#### 2.4.4. Performed Analysis

Behavioral, subjective, and neurophysiological signals collected during each scenario have been analyzed to assess the drivers’ workload and so to test the capability to measure the drivers’ mental improvements during a training program, such as learning to drive. 

In terms of neurophysiological measures, the mental effort (ME) was employed, as described in Section 2.4.1. The reliability of the ME neurometric was initially tested on the labelled conditions, i.e., *Easy* and *Hard* driving tasks, in order to prove its sensitivity to the difficulty of the task. Then, it was used to independently assess the mental effort experienced by the driving trainees during the 5 repetitions (*Runs*) of the urban driving task. Behavioral and subjective measures collected during the 5 *Runs* were analyzed accordingly.

While comparing two conditions, the Shapiro–Wilk test of normality was preliminary performed on the data distribution [66], since the kind of distribution impact on the statistical analysis to perform: if data are normally distributed, parametric analysis can be performed, otherwise a non-parametric analysis is preferable when it is not possible to demonstrate data normality. 

Therefore, when comparing two conditions, a two-tails paired Student’s t-test was performed in case of normality confirmation, otherwise a Wilcoxon.

While comparing more conditions, a repeated measures ANOVA, and eventually a Duncan’s post hoc test, was performed, or a Friedmann nonparametric test, and related Conover’s post hoc test. For all the statistical tests, a threshold value of *p* = 0.05 was considered, that means a statistically significant result when *p* < 0.05.

## 3. Results

### 3.1. Subjective Assessment

Here, the analysis of the answers provided to the NASA-TLX questionnaire, in particular in terms of ‘Effort’, ‘Performance’, and overall ‘Workload’. None of the analysis shows any significant effect:(1)Effort: F (4, 68) = 1.232; *p* = 0.306;(2)Performance: F (4, 68) = 2.224; *p* = 0.076;(3)Workload: F (4, 68) = 0.195; *p* = 0.940.

No significant effects were found on subjective measures. If considering the difference between the first and last run, a decreasing trend seems to appear from the Effort assessment, while an increasing trend is shown by the assessment of performance satisfaction. Moreover, the overall workload does not show any trend, probably because of the concomitant effect of the different factors. In conclusion, besides a mere visual effect, it is not possible to assess any improvement by analyzing self-assessments.

### 3.2. Behavioural Assessment

In terms of driving performance, no significant effects arose from the analysis of the number of collisions while driving (Friedman test: χ^2^ (4) = 0.393; *p* = 0.983).

On the contrary, the analysis of reaction times to *failures* showed a significant effect in terms of reaction times decreasing (Friedman test: χ^2^ (4) = 19.067; *p* < 0.001).

As hypothesized, with the practice the reaction times decreased significantly, thus suggesting that the participants were able to perform the driving task by allocating less cognitive resources, and thus becoming more reactive towards external events. Post hoc test showed that the reaction times decreased up to the third repetition (*Run 3*), then no significant differences were found. 

### 3.3. Neurophysiological Assessment

Firstly, the capability of the ME neurometric, computed as the GFP in theta band over the frontal channels, in discriminating two driving tasks different in terms of difficulty was tested over the *Easy* and *Hard* driving tasks.

The statistical analysis confirmed this assumption, since the paired Student’s *t*-test (the distributions normality was verified, Shapiro–Wilk test: *p* = 0.343) showed that the EEG-based mental effort scores during the *Hard* task were significantly higher than during the *Easy* task (*t* = −2.807; *p* = 0.011; Cohen’s d = −0.613).

Therefore, the EEG-based mental effort scores have been used to investigate the mental abilities of the participants while performing the five repetitions of urban driving tasks. Statistical analysis showed a significant effect in terms of effort decreasing across the runs (F (4, 68) = 2.536; *p* = 0.0479).

The analysis of mental effort highlighted an actual decrease in the mental effort experienced by the drivers across the driving tasks repetition. In particular, from the third repetition the effort was significantly lower than the first one.

## 4. Discussion

During a training program, it is usually difficult to properly assess the trainees’ progresses, since instructors and external evaluators can only rely on the evaluation of the explicit behavior of the trainee and on eventual performance indicators. Moreover, the self-evaluation does not solve the question, since it is usually hard to be fully aware of one’s own psychophysiological state. 

The present study aimed to deal with this concern, by investigating the deployment of wearable EEG technology in order to objectively assess the trainee’s performance improvements, while repeating a similar driving task several times, from a cognitive point of view.

Traditional techniques, such as the trainee’s self-assessment and the analysis of driving performance, in terms of collisions, were demonstrated to be not sensitive enough, as hypothesized. In fact, from one side, the trainees were not able to perceive any significant improvement from the first to the fifth repetition of the driving task (Figure 5).

Apart from a trend in terms of performance satisfaction (*p* = 0.076), no significant results were found in terms of cognitive improvements, such as the perceived mental effort and the overall workload. In other words, it is not possible to assess any improvement by analyzing self-assessments. It is not surprising not to have obtained significant evidence from subjective and behavioral data; in fact it is well accepted in the literature that subjective and behavioral measurements are more affected by individual bias and confounds, thus resulting in “noisy measures”, so large numbers in terms of sample sizes are needed, while, on the other hand, neurophysiological measures, being more “objective”, allows to use fewer data to obtain significant evidence [67].

At the same time, no evidence arose from the analysis of driving performance (Figure 6), i.e., what is generally visible to an external supervisor and/or trainer. It has to be noted that this measurement was probably affected by the short duration of the driving tasks, which resulted in a low number of collisions from the first run. Perhaps longer tasks would have increased the probability of collisions during the first repetitions and thus favored the sensitivity of this indicator across them.

On the other side, the behavioral analysis in terms of reaction times on a secondary ecological task (switching off a car alarm) highlighted a strong effect (Figure 7), with a significant improvement over time (*p* < 0.001). In particular, post hoc analysis revealed a significant decreasing (lower means faster thus better RTs) until the third run, without any further improvement after that). In general, RTs on secondary tasks are directly linked to the mental spare capacity [68,69,70]: assuming that human cognitive resources are not infinite, the less demanding the main task is, the more cognitive resources will remain available, thus allowing improvements also on secondary tasks [71]. Therefore, this result would suggest that the main task, i.e., driving the car, is becoming easier and easier for the trainee.

Coherently, through the analysis of the drivers’ mental activity, i.e., the EEG-based neurometric of mental effort, preliminarily validated in terms of sensitivity in discriminating two driving tasks with different difficulty (Figure 8), it was possible to unveil trainees’ improvement in terms of mental performance. In fact, the results showed a significant decreasing (*p* = 0.0479) of mental effort (Figure 9), that from the third run became significantly lower than the first one (post hoc analysis). 

This result is extremely relevant because, apart from demonstrating the added value of such a neuroergonomics approach with respect to traditional ones in assessing the learning curve of a trainee, it shed light on a critical concern in the automotive field. In fact, driving a vehicle is always the main but very rarely the only task we perform on the road [72]: while driving, we have to perform a huge number of secondary tasks, first of all monitoring the surroundings, predicting the behavior of other vehicles and pedestrians, identifying and anticipating possible hazards, monitoring the vehicle’s diagnostics, managing the infotainment and air conditioning system, etc., not to mention the additional sources of distraction such as passengers, phone calls, and messages. Therefore, being able to drive properly and safely does not mean simply to know how to practically conduct the vehicle, but to be able to do it with a coherent amount of cognitive resources in order not to lose situation awareness and to remain able to react to sudden and unexpected events [73,74,75].

This neuroergonomic approach would improve driving educational services from different perspectives: the main one is that directly suggested by the present work, i.e., to understand whether the student is ready to move forward along the training program. The result would be an individualized driving training program specifically tailored on the trainee’s skills and needs. However, besides this immediate impact, such a technology would also enable the possibility of performing automatic evaluation of mental skills, for example through the inclusion of intermediate practical tests that can be performed in a simulator instead of a real car. Of course, a similar impact would need a huge set of collateral concerns, such as benchmark definitions and proper legislation, discussion of which is outside the scope of the present work.

However, it is important to point out certain limitations that affect the study. First of all, the experimental sample, which is strongly unbalanced in gender. In any case, all statistical analyses are paired or repeated measures, so the influence of the individual subject is minor. Moreover, the sample size could affect the study robustness. To this regard, a recent study [31] demonstrated that in long protocols (measurements longer than 30 s) sample sizes around 22–24 participants are enough to obtain results highly correlated with results obtained from a population of 36 participants. In any case, study replication with a larger sample size is encouraged. Furthermore, the driving tasks used are very short. Finally, other neurophysiological measures could be employed to obtain a more holistic assessment of the student’s psychophysiological performance, e.g., by investigating other psychocognitive concepts such as stress and attention, which are equally important in this context.

In any case, the study remains relevant and novel with respect to the current state-of-the-art, demonstrating the potential benefits of integrating these modern neuroscientific tools in social applications such as driving education, where they can be really game-changing, opening new frontiers on the evaluation of the trainee in terms of both practical (traditional approach) and cognitive (neuroergonomics) skills and performance, towards a better and more effective education.

## 5. Conclusions

The present study investigates the possibility of including the EEG technology along a driving education program for a better assessment of trainee’s cognitive improvements. Actually, the EEG-based neurometric of mental effort highlighted the significant decrease in cognitive demand while repeating the driving tasks, concomitantly with the significant improvement of its reaction times on a secondary task. From subjective (self-assessment) and behavioral (driving performance) measures it was not possible to obtain any significant evidence. This result conveys the sense and the added value of a neuroscientific approach focused on users’ cognitive experience: being able to measure when a given task is more automatic and therefore less demanding for the user, allows to deduce that users will then be more capable of handling additional tasks and reacting to unexpected events.

## Figures and Tables

**Figure 1 sensors-23-08389-f001:**
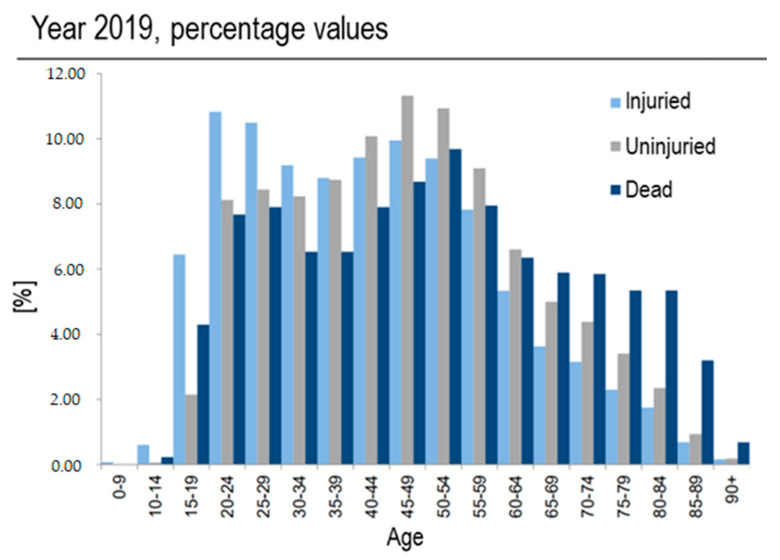
Percentage (y axis) of drivers involved in road accidents, divided per age (x axis) and consequences (color: light blue for ‘Injured’, grey for ‘Uninjured’, dark blue for ‘Dead’). Source: ISTAT Report 2019.

**Figure 2 sensors-23-08389-f002:**
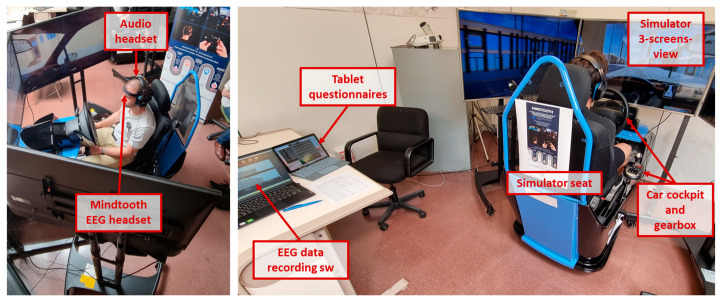
Experimental room at the University of Burgos with the two car simulators. The participant, seated in the simulator, wears the audio headset and the Mindtooth EEG system. The EEG data were collected by means of the Mindtooth proprietary recording software (v.5.22). The simulator consists of a real car seat, with a coherent cockpit, the steering wheel, and the gearbox. A tablet was used to provide self-assessment questionnaires digitally.

**Figure 3 sensors-23-08389-f003:**
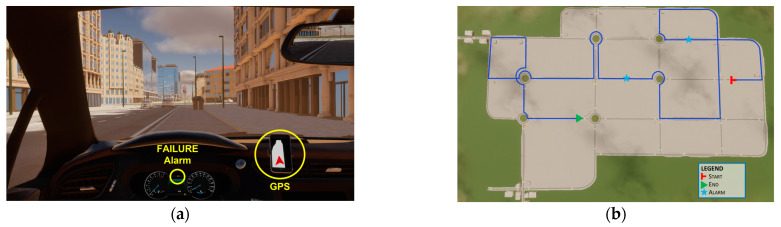
On the left (**a**), a screenshot of the simulated driving scenario within the urban environment: highlighted the failure alarm light on the dashboard and the GPS system indicating the route to follow. On the right (**b**), the route (in blue) designed for the adaptation task with the position of the failure events (‘ALARM’).

**Figure 4 sensors-23-08389-f004:**
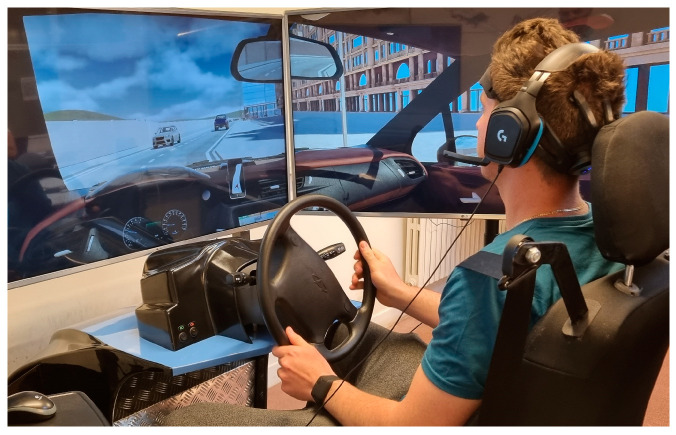
A picture of one participant while performing an urban driving task.

**Figure 5 sensors-23-08389-f005:**
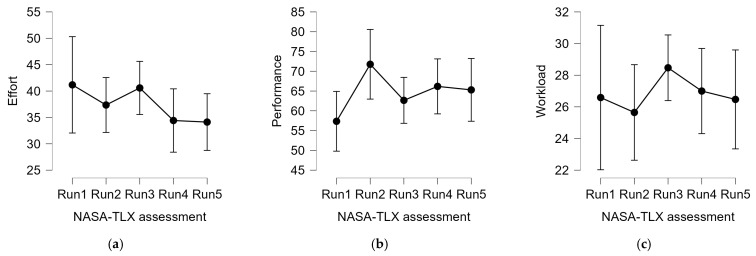
Analysis of the distribution of scores (mean ± 95% confidence interval) self-assessed by the participants with respect to the 5 runs of urban driving tasks in terms of, from the left to the right, (**a**) Effort, (**b**) Performance, and (**c**) overall Workload (as the mean of all the 6 NASA-TLX dimensions.

**Figure 6 sensors-23-08389-f006:**
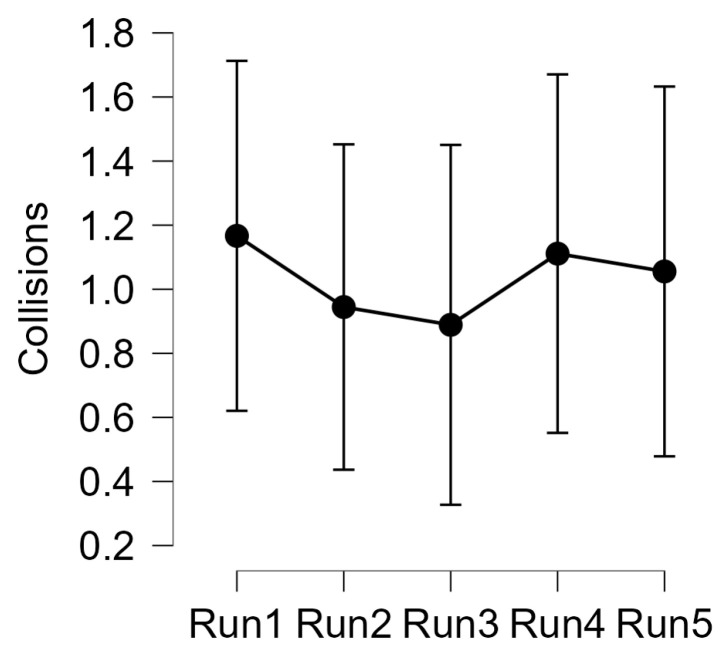
Analysis of the distribution of collisions while driving (mean ± 95% confidence interval) with respect to the 5 runs of urban driving tasks.

**Figure 7 sensors-23-08389-f007:**
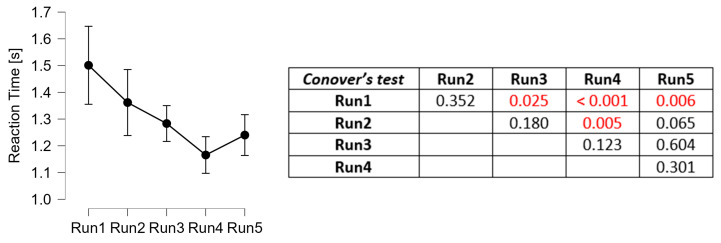
Analysis of the reaction times (mean ± 95% confidence interval) with respect to the 5 runs of urban driving tasks. On the right, the *p*-values of the paired comparisons with the Conover’s post hoc test (in red, the statistically significant values—*p* < 0.005).

**Figure 8 sensors-23-08389-f008:**
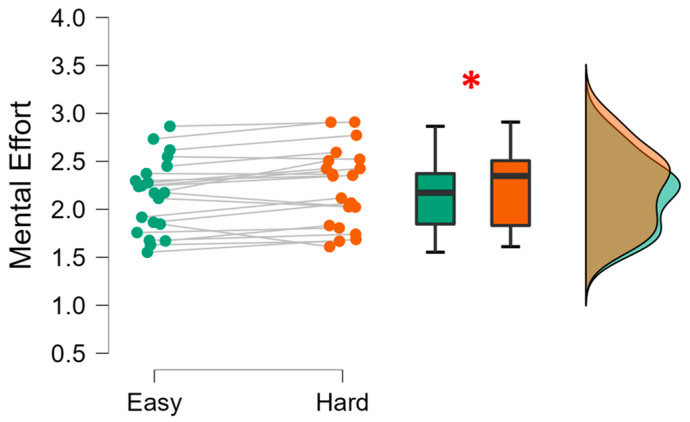
Analysis of the distribution of EEG-based workload scores with respect to the Easy (green) and Hard (orange) driving tasks. The red asterisk indicates the statistically significant effect.

**Figure 9 sensors-23-08389-f009:**
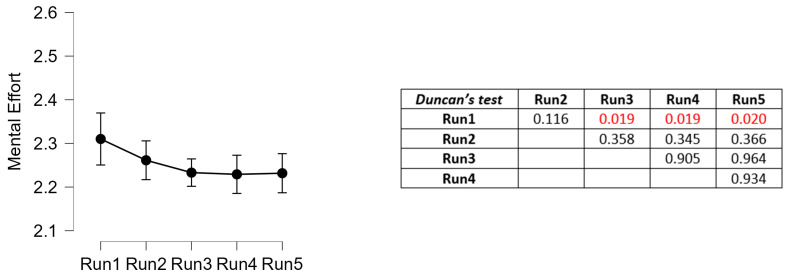
Analysis of the EEG-based mental effort scores (mean ± 95% confidence interval) with respect to the 5 runs of urban driving tasks. On the right, the *p*-values of the paired comparisons with the Duncan’s post hoc test (in red, the statistically significant values—*p* < 0.05).

**Table 1 sensors-23-08389-t001:** Experimental protocol, with the list of tasks in a chronological order. The five driving tasks indicated by the asterisk consisted of five different routes. In blue, the questionnaires provided at the end of each experimental task.

**“Driving Trainees Training” Protocol** *Total Duration ≈ 90 min*
1. Adaptation driving task 2. Open eyes condition 3. Closed eyes condition 4. *Hard* driving task 5. *Easy* driving task 6. Urban driving task*—*Run1* 7. NASA-TLX 8. Urban driving task*—*Run2* 9. NASA-TLX 10. Urban driving task*—*Run3* 11. NASA-TLX 12. Urban driving task*—*Run4* 13. NASA-TLX 14. Urban driving task*—*Run5* 15. NASA-TLX

## Data Availability

The aggregated data analyzed in this study might be available on request from the corresponding author.

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
