# Peer review of "A Neuroergonomic Approach Fostered by Wearable EEG for the Multimodal Assessment of Drivers Trainees"

_sensors, 2023, doi:10.3390/s23208389_

Round 1

Reviewer 1 Report

 ** General comments

The study theoretically proposes an assessment of mental effort through EEG to get a quantitative metric for drivers trainees that is more significant than subjective and behavioral assessment. However, the study has some flaws and aspects to be improved. As a main one, although the abstract, introduction and conclusions affirm that the EEG-related analysis is the core of the study, the methods and results appear to give more relevance to the experimental setup and protocol. Specific comments are reported in the following in hoping to help the Authors with necessary improvement. Overall, the structure of the manuscript and English drafting must be improved.

 ** Major comments

1.      what is a "neurometric"?

2.      The proposal, as well as the experimental protocol, is unclear from the abstract, therefore it is not possible to fully understand what a "decrease of experienced mental demand from the third run" implies

3.      From the abstract the overall idea is not clear, and it is already highlighted that English drafting must be improved as well

4.      please be aware that the graph in Figure 1 has a title and legend in Italian!

5.      The need to consider driving education as a major social issue does not appear as properly motivated with literature or other sources, but the reasoning appears as rather subjective

6.      there are EEG-related works mentioned in the introduction, but they are not discussed and even more a comment about already existing literature for methods inherent with the proposal is missing, so that the reader cannot evaluate the novelty and significance of the work. In improving this part, I would suggest to also have a look at https://doi.org/10.1109/ACCESS.2022.3219844

7.      at the end of the introduction, it would be useful to add a recap of the structure of the manuscript

8.      the materials and methods section should start with the proposal before describing the experiments. Hence, it could make sense to describe the experimental tasks (although I would not put them as first), but it does not help to describe the experimental sample and protocol before detailing the proposal. By adjusting that, the contribution of the work could be appreciated

9.      The quality of Fig. 3b should be definitely improved since it is half-handwritten

10.   The assumptions implying the choices for the statistical tests should be reported and clarified. moreover, in Section 2, the threshold p-value chosen for the tests is not reported, while that is essential for interpreting test results

11.   the Authors interpret the results of statistical tests in a slightly wrong way: when there is no significant effect, the test simply implies that "there is not enough evidence to reject the (null) hypothesis" but this is fundamentally different from saying that the alternative hypothesis can be accepted! This especially applies to the normality test in the section 3, while in other parts this is reported correctly

12.   given the error bars, one cannot really speak of a decreasing trend for the "Effort assessment"

13.   "... the study remains extremely relevant" appears as a rather subjective statement which should be avoided in a scientific text

** Minor comments

1)     In the abstract, I would avoid the first paragraph with background because it is better to put it in the introduction and go straight to the proposal in the abstract

2)     instead of "very few driving experience", maybe it would be better to say "limited driving experience"

3)     upper case letters appear useless in "Mental Effort". The same is true in many other parts, e.g. in “Easy”/”Hard”, “Open”/”Closed Eyes” and so on

4)     there is a missing space before references [1,2]. the same is true for other references

5)     it should be "still be not ready" instead of "still not be ready"

6)     if an abbreviation is never used again, it should not be introduced (see e.g. EOG). Moreover, the number of abbreviations should be limited because they save little ink and make everything unclear

7)     The "Figure 3" is not referenced nor explicitly explained in the manuscript body

8)     the notation "90 degrees" is not compliant with the SI of units, and the notation "2:30 minutes" is not compliant with the SI of units either

9)     It is unnecessary to put (7) after "Seven"

10)  the upper case letters for RUN and FAILURE appear a little confusing because they could resemble abbreviations. I would rather use bold or italics for those events

11)  Starting a paragraph (or even a sentence) with a number like in "24 healthy..." may be inappropriate

12)  the number of decimal digits in 19.2 +/- 1.18 is incoherent, it should instead be 19.2 +/- 1.2 or it could be also 19 +/- 1

13)  please use the symbol for the "ohm" measuring unit rather than kOhm (which is not compliant with SI for different reasons)

14)  please pay attention to spacing before paragraphs, section, subsections and also before references like "Par.2.4.1" (there is a missing space)

15)  please pay attention to harmonize names and notations: i.e., HARD and EASY may appear different from "Hard" and "Easy"

16)  with the "F" of "Filter" is an upper case letter but the "multi-channel" has no upper case letter?

17)  why "Threshold"?

18)  at page 7, line 259: "was demonstrated" should be preferred as a verbal tense since that thing was ALREADY demonstrated in the past. Please harmonize the verbal tenses in general

19)  at page 8, line 278: there should be no indent and no upper case letter for "where"

20)  also the Theta band is named in different ways and this appears confusing

21)  page 8, line 281: it should be "1 s"

22)  pay attention to the caption of Fig. 5 which is in a page different from the one of the actual figure

23)  "Reaction to failures" does not appear correct as x-axis label for Fig. 7, and also the label "EEG Neurometric" for the x-axis in Fig. 9 does not seem correct

The overall English drafting should be improved: please refer to the main comments for more punctual examples of aspects to improve

Author Response

We thank the reviewer for its comments and suggestions, we took them into account in order to improve the manuscript. Overall, a careful English editing has been performed. In addition, here attached our point-by-point answers, with reference to the modifications in the manuscript (the lines are those in the track-changes document, i.e. the pdf one).

Reviewer 2 Report

1.The lack of detailed information to reproduce the study, as well as the insufficient reporting of data analysis techniques. 

It is worth highlighting in your Limitations  that why there  was not possible to get any significant evidence from subjective rating and driving performance.

2.Please elaborate more if a Power analysis was performed? 22 in each group may be a small group size. Why weren’t more students recruited to reach at least 30 in each group?

The justification for the chosen sample size should be provided, including the results of the power analysis.

3.The discussion section of the manuscript requires further elaboration. Currently, the manuscript primarily presents a summary of the results, lacking a comprehensive analysis of their impact on training.

The manuscript read like it was written by more than one author. There was not a consistent voice / style.

Author Response

We thank the reviewer for its comments and suggestions, we took them into account in order to improve the manuscript. Here attached our point-by-point answers, with reference to the modifications in the manuscript (the lines are those in the track-changes document, i.e. the pdf one).

Round 2

Reviewer 1 Report

Most of the comments from the previous round were addressed in a quite satisfying way, but some others were not (and other new things to adjust also appeared). Here is a point-to-point list of issues to address, where the numbering recalls the one of the previous round for clarity. The biggest remaining issue in my opinion is in the not-very-strong results.

Major comment 1 – the explanation of a “neurometric” added in the text is quite satisfying, but I do not agree with the usage of the upper case letter

Major comment 2 and 3 – the abstract is now clearer, but please note that from the cover letter it is unclear which parts were actually edited, while explicitly indicating that would have sped up the review process.

Major comment 5 - good, properly motivated now, although I would have mentioned the discussion with the stakeholders more explicitly

Major comment 8 – I still do not agree with this structuring of the paper

Major comments 10 to 12 – these comments appear as poorly addressed in the new version of the manuscript, and since the manuscript did not change much, the results still appear rather weak. Specifically, I need to remark that one cannot speak of a decrease in effort, even when considering the only first and last run, because the two intervals are compatible!!

Minor comment 3 - I am still in disagree with that because I would use italics to emphasize, and I do not think upper case letters are useful for Electroncephalographic and similar terms

Minor comment 6 – it is not true that abbreviations like EOG were avoided…

Minor comment 8 - "90 degrees" still remains in line 213, and also "minutes" still appears instead of "min", while in other parts “second” appears instead of “s”, and so on…

Minor comment 10 – also “RUN” is still appearing

Minor comment 13 – the issue was fixed, but now a “kilo-ohm” appears instead of kW

Minor comment 17 – please apologize for the lack of clarity in the previous round, but I was referring to the use of and upper case letter for “Threshold”. In general, as it could be noted also before, I do not agree with the usage of upper case letters in many other parts

Minor comment 21 – a new notation was introduced that it needs to be harmonized: 1-s-long vs 1-second-long

I still do not fully agree with the usage of present perfect as a verbal tense, in English it is more common to use past tenses for already done things. Moreover, as reported above, I do not agree with the usage of upper case letters

Reviewer 2 Report

It is better to redraw the Figure 2 to show the layout of the two car simulators.

 I recommend at the very least using a free online tool to help you edit and catch errors.

Author Response

Thank you for the suggestion, the figure has been edited accordingly.